# Epigallocatechin-3-Gallate Suppresses Vasculogenic Mimicry through Inhibiting the Twist/VE-Cadherin/AKT Pathway in Human Prostate Cancer PC-3 Cells

**DOI:** 10.3390/ijms21020439

**Published:** 2020-01-09

**Authors:** Changhwan Yeo, Deok-Soo Han, Hyo-Jeong Lee, Eun-Ok Lee

**Affiliations:** 1Department of Cancer Preventive Material Development, College of Korean Medicine, Graduate School, Kyung Hee University, 26, Kyungheedae-ro, Dongdaemun-gu, Seoul 02447, Korea; duelf2@naver.com (C.Y.); strong70@khu.ac.kr (H.-J.L.); 2Department of Science in Korean Medicine, College of Korean Medicine, Graduate School, Kyung Hee University, 26, Kyungheedae-ro, Dongdaemun-gu, Seoul 02447, Korea; ejr0957@khu.ac.kr

**Keywords:** EGCG, vasculogenic mimicry, twist, VE-cadherin, AKT, PC-3 cells

## Abstract

Vasculogenic mimicry (VM) is the alternative process of forming vessel-like networks by aggressive tumor cells, and it has an important role in tumor survival, growth, and metastasis. Epigallocatechin-3-gallate (EGCG) is well known to have diverse bioactivities including anti-cancer effects. However, the efficacy of EGCG on VM is elusive. In this study, we explored whether and how EGCG affects VM in human prostate cancer (PCa) PC-3 cells. Cell viability was measured by 3-(4,5-dimethylthiazol-2-yl)-2,5-diphenyltetrazolium bromide (MTT) assay. Invasive and VM formation abilities were assessed by an invasion assay and a three-dimensional (3D) culture VM tube formation assay, respectively. Western blots were carried out. An immunofluorescence assay was performed to detect nuclear twist expression. EGCG effectively inhibited the invasive ability, as well as tubular channel formation, without affecting cell viability. EGCG significantly downregulated the expression of vascular endothelial cadherin (VE-cadherin) and its transcription factor, twist, N-cadherin, vimentin, phosphor-AKT, and AKT, but not phospho-erythropoietin-producing hepatocellular receptor A2 (EphA2) and EphA2. In addition, EGCG diminished the nuclear localization of twist. Treatment with SC79, an AKT activator, effectively rescued EGCG-inhibited VM formation. These results demonstrated for the first time that EGCG causes marked suppression of VM through inhibiting the twist/VE-cadherin/AKT pathway in human PCa PC-3 cells.

## 1. Introduction

Blood supply is required to promote tumor growth, survival, and metastasis [1]. Angiogenesis was first proposed in 1971 by Judah Folkman as new blood vessel formation from pre-existing ones by endothelial cells [2,3]. Because angiogenesis plays important roles in tumor growth, survival, and metastasis, anti-angiogenic therapies were considered as promising and key strategies for the treatment of cancer for almost 30 years [2,4]. However, anti-angiogenic drugs were not effective in all cancers, and resistance to these therapies occurred during a period of treatment [5]. Animal studies also reported that anti-angiogenic therapy delays tumor growth, followed by tumor regrowth [6]. These researches demonstrated that tumor cells can evade the therapeutic effects of anti-angiogenic drugs and suggested that these phenomena may be due to adequate blood supply via alternative pathways.

Maniotis et al. [7] reported a novel tumor vascular paradigm called “vasculogenic mimicry (VM)” in 1999. VM is the de novo formation of perfusable, matrix-rich, and vessel-like networks by aggressive tumor cells without endothelial cells, and it provides sufficient oxygen and nutrients for tumor survival and growth through direct or indirect connections with endothelial-lined vessels [8,9,10]. Various genes such as vascular endothelial cadherin (VE-cadherin) and erythropoietin-producing hepatocellular receptor A2 (EphA2), and signaling pathways including the phosphoinositide 3-kinase/AKT pathway participate in VM formation by aggressive tumor cells [9,11]. VM formation was observed in various kinds of tumors including prostate cancer (PCa), and it is strongly involved in poor clinical outcome [12,13]. Overall survival and disease-free survival of patients with VM-positive PCa are significantly worse than those with VM-negative PCa [12]. In a meta-analysis, VM-positive cancer patients such as those with lung, colon, and liver cancers showed a poor five-year survival [13]. These results indicated that VM is an adverse predictor in cancer patients. Food and Drug Administration (FDA)-approved anti-angiogenic inhibitors such as bevacizumab, endostatin, and TNP-470 fail to inhibit tumor VM formation [9,11,14]. Therefore, VM is an attractive target that can improve anti-cancer efficacy through overcoming resistance to anti-angiogenic therapies. In addition, the combination of VM-targeting drugs and anti-angiogenic therapies may have a synergistic effect in the treatment of vascular-related cancer patients such as PCa.

Green tea is one of the most popular beverages consumed around the world due to it containing powerful antioxidants such as polyphenols [15]. Epigallocatechin-3-gallate (EGCG) is the most abundant polyphenol in green tea [16]. It showed potential efficacy on human health and diseases such as cancer, also in addition to cardiovascular, metabolic, and neurodegenerative diseases [16,17]. EGCG possesses anti-cancer capacities through the induction of apoptosis [18,19] and the repression of angiogenesis, metastasis, and tumor growth [20,21]. Also, EGCG acts as a chemosensitizer, leading to minimizing chemoresistance and enhancing chemosensitivity of tumor cells [22]. However, there are few researches on EGCG’s effect on VM. Only one study showed that EGCG blocked VM formation through the reduction of intracellular peroxide [23].

Thus, the aims of this study were to determine the effects of EGCG on VM formation and to identify the underlying molecular mechanism of its activity in human PCa PC-3 cells.

## 2. Results

### 2.1. The Effect of EGCG on the Viability of PC-3 Cells

To evaluate the effect of EGCG (Figure 1A) on the viability of PC-3 cells, we treated PC-3 cells with various concentrations of EGCG (10, 20, 40, and 80 μM) for 24 h and performed a 3-(4,5-dimethylthiazol-2-yl)-2,5-diphenyltetrazolium bromide (MTT) assay. Here, 80 μM EGCG significantly decreased cell viability. However, there was no effect of EGCG on cell viability up to 40 μM (Figure 1B). Therefore, to rule out the possibility of inhibiting VM by cytotoxicity, we treated PC-3 cells with noncytotoxic concentrations (≤40 μM) of EGCG for subsequent experiments.

### 2.2. EGCG Reduces the Invasion of PC-3 Cells

To check the anti-invasive activity of EGCG against PC-3 cells, we conducted a cell invasion assay using a Transwell with matrigel-coated membrane filter for 24 h. Fetal bovine serum was used as a chemoattractant. As expected, 10% serum caused a marked increase in cell invasion ability, which was effectively reduced by 25%, 38%, and 62% with the 10, 20, and 40 μM EGCG treatments, respectively (Figure 2). These results verified that EGCG has an anti-invasive activity in PCa PC-3 cells.

### 2.3. EGCG Inhibits the VM of PC-3 and DU-145 Cells

To investigate whether EGCG affects the formation of vessel-like networks by PCa such as PC-3 and DU-145 cells, we treated the cells on the matrigel-coated wells with EGCG and then carried out a three-dimensional (3D) culture VM tube formation assay for 24 h. As shown in Figure 3A, PC-3 cells formed complete tubular channels, which was partly blocked by EGCG treatment. VM tube formation of PC-3 cells was dramatically inhibited by 15%, 31%, and 57% with 10, 20, and 40 μM EGCG, respectively (Figure 3B). Also, EGCG effectively reduced the VM formation of DU-145 cells by 20%, 36%, and 67% with 10, 20, and 40 μM, respectively (Figure 3C,D). These results demonstrated that EGCG has an anti-VM activity in PCa cells.

### 2.4. EGCG Downregulates VE-Cadherin Expression through Inhibiting the Nuclear Twist in PC-3 Cells

To examine the role of EGCG on EphA2 phosphorylation and VE-cadherin expression involved in VM formation, we analyzed the protein levels of these key factors by Western blot in EGCG-treated PC-3 cells for 24 h. There was no significant difference in phospho-EphA2 or EphA2 expression (data not shown). However, VE-cadherin expression was strikingly downregulated by ECGC treatment in a dose-dependent manner (Figure 4A). These results revealed that the downregulation of VE-cadherin but not EphA2 is associated with EGCG-inhibited VM formation in PC-3 cells.

Next, to investigate whether EGCG affects twist expression to inhibit VM formation through regulating VE-cadherin in PC-3 cells, we detected the protein expression level and the nuclear localization of twist by Western blot and immunofluorescence assay after EGCG treatment for 24 h. As shown in Figure 4B, EGCG significantly reduced the twist expression level in a dose-dependent manner. The nuclear localization of twist was effectively diminished by 20 μM EGCG (Figure 4C). To check the transcriptional regulation of twist target genes such as N-cadherin and vimentin, Western blot was performed after EGCG treatment for 24 h. As expected, N-cadherin and vimentin expression was downregulated by EGCG in a dose-dependent manner (Figure 4D).

Taken together, these results indicated that EGCG inhibits twist level in the nucleus, thereby downregulating VE-cadherin expression in PC-3 cells.

### 2.5. EGCG Suppresses the VM of PC-3 Cells through Inhibiting the AKT Pathway

To verify the association between EGCG and the AKT pathway, Western blot was carried out in EGCG-treated PC-3 cells for 24 h. EGCG significantly impaired phospho-AKT and AKT expression level in a dose- and time-dependent manner (Figure 5A,B). To assess the role of the AKT pathway in EGCG-inhibited VM, we treated PC-3 cells with SC79, an AKT activator, in the presence or absence of EGCG and then performed a 3D culture VM tube formation assay. Only SC79 treatment had no effect on VM formation. However, EGCG significantly resulted in blockage of VM formation, which was rescued after SC79 treatment (Figure 5C,D). These results demonstrated that EGCG requires inhibiting the AKT pathway to suppress VM formation in PC-3 cells.

## 3. Discussion

VM is the alternative vessel formation by aggressive tumor cells without endothelial cells. It contributes to blood supply for sufficient oxygen and nutrients for tumor survival and growth [7,8,9,10]. The patients with VM-positive PCa have poorer overall survival and disease-free survival than those with VM-negative PCa [12]. Also, in a meta-analysis, VM-positive cancer patients such as those with lung, colon, and liver cancers showed a poor five-year survival [13]. Therefore, VM acts as a marker of poor prognosis in various kinds of tumors including PCa [12,13], indicating that VM is an attractive target for treating cancer patients.

The invasive ability of tumor cells is closely associated with VM [24]. EGCG was reported to inhibit the invasion of several types of cancer cells [25,26,27]. As expected, EGCG clearly inhibited the serum-induced invasion of PC-3 cells (Figure 2). Also, VM formation by PC-3 and DU-145 cells was dramatically blocked after EGCG treatment (Figure 3). All these results were assessed at noncytotoxic concentrations (≤40 μM) of EGCG (Figure 1B). Taken together, these results demonstrated that EGCG has anti-invasive and anti-VM abilities without affecting cell viability.

Vascular-associated genes such as VE-cadherin and EphA2 are involved in the ability of aggressive tumor cells to form VM [9,11]. As one of the first vascular-associated genes, VE-cadherin is a member of classical cadherin adhesion molecules, and it engages in blood vessel formation through controlling endothelial cell behavior [9,28]. EphA2 belongs to the ephrin-receptor family of receptor tyrosine kinases, and it regulates cancer motility, proliferation, and stemness properties by phosphorylation [29]. VE-cadherin and EphA2 are overexpressed in aggressive cancer cells and play key roles in the formation of matrix-rich vessel-like networks called VM. Knockdown of not only EphA2 but also VE-cadherin abrogated VM formation in aggressive melanoma cells [29,30]. In this study, EGCG downregulated VE-cadherin expression (Figure 4A) but not phospho-EphA2 and total EphA2 expression (data not shown). These results verified that EGCG could inhibit VM formation through the downregulation of VE-cadherin expression.

As a transcription factor, twist induces VM through upregulation of VE-cadherin [31,32]. Twist is overexpressed in the nucleus in VM-positive hepatocellular carcinoma (HCC). The survival of HCC patients with positive VM, nuclear twist, and VE-cadherin expression is shorter than that of those without expression [31]. Blocking the twist/VE-cadherin pathway contributes to the suppression of VM formation [33]. Among PCa cell lines, PC-3 cells shows a high expression of twist [34]. As expected, EGCG strikingly reduced twist expression (Figure 4B) and the nuclear localization of twist (Figure 4C). Epithelial–mesenchymal transition (EMT) is a vital process in cancer progression, and it also plays a crucial role in VM formation [35,36]. Twist is a major EMT-related transcription factor and controls EMT through downregulating epithelial markers (E-cadherin) and upregulating mesenchymal markers (N-cadherin and vimentin) [36]. Consistent with the results of VE-cadherin and twist, EGCG downregulated N-cadherin and vimentin (Figure 4D).

Taken together, EGCG downregulated VE-cadherin expression through inhibiting the nuclear localization and expression of twist, leading to suppressing VM formation.

The AKT pathway plays an important role in cancer progression, related to cell survival, growth, angiogenesis, and metastasis [37]. Also, the AKT pathway contributes to VM formation. VE-cadherin activates the AKT pathway, resulting in VM formation through controlling the activity of matrix metalloproteinase and the cleavage of laminin subunit 5 gamma-2 [8,11]. Curcumin inhibits VM of HCC cells through downregulating the AKT pathway [38]. EGCG reduced phospho-AKT and AKT expression (Figure 5A,B). The EGCG-induced inhibition of VM formation was blocked by an AKT activator (Figure 5C,D). These results demonstrated that impairing the AKT pathway contributes to the anti-VM ability of EGCG.

In summary, we demonstrated that EGCG could suppress the VM formation of PCa PC-3 cells through inhibiting the nuclear localization of twist, followed by downregulation of VE-cadherin expression, which in turn impairs the AKT pathway. These results serve new insight into the function of EGCG in VM and suggest that more research is needed to develop EGCG as a chemopreventive and chemotherapeutic agent targeting VM.

## 4. Materials and Methods

### 4.1. Cell Culture

Human prostate cancer cell lines PC-3 and DU-145 were obtained from the Korea Cell Line Bank (KCLB, Seoul, Korea). All cells were maintained in RPMI 1640 with 10% fetal bovine serum (FBS, Welgene, Daegue, Korea) and 1% antibiotics in a humidified atmosphere of 5% CO_2_ at 37 °C.

### 4.2. Cell Viability Assay

Cells (1 × 10^4^) were seeded in a 96-well plate, incubated for 24 h, and then treated with various concentrations (10, 20, 40, and 80 μM) of EGCG (Enzo Life Sciences, Farmingdale, NY, Purity ≥98% by HPLC) in serum-free media for 24 h. The effect of EGCG on the viability of PC-3 cells was measured by a 3-(4,5-dimethylthiazol-2-yl)-2,5-diphenyltetrazolium bromide (MTT, Sigma-Aldrich, St Louis, MO, USA) assay as described previously [39,40,41]. The absorbance at 570 nm was measured using a microplate reader (Sunrise RC, TECAN, Mannedorf, Switzerland). Cell viability was expressed as a percentage of the control.

### 4.3. Invasion Assay

The cell invasion assay was performed using Transwell^®^ cell culture inserts with 8-μm pores (Corning Inc., NY, USA). The membrane filter of an insert was precoated with dilute matrigel (1:20, BD Biosciences) for 2 h at 37 °C. The lower chamber was filled with 700 μL of RPMI 1640 containing 10% FBS (Welgene), and 400 μL of a cell suspension (2 × 10^5^/well) with various concentrations (10, 20, and 40 μM) of EGCG in serum-free media was placed in the inserts. After incubation for 24 h at 37 °C, the inserts were fixed and stained with Diff-Quick (Sysmex, Kobe, Japan), and the cells on the upper surface of the membrane filter were removed with a cotton swab. Then, the invaded cells were captured using an inverted light microscope Ts2_PH at 200× magnification (Nikon, Tokyo, Japan) and quantified.

### 4.4. Three-dimensional (3D) Culture VM Tube Formation Assay

A 24-well plate was added with 100 μL/well of matrigel basement membrane matrix (BD Biosciences, San Jose, CA), and then allowed to polymerize for 1 h at 37 °C. A cell suspension (PC-3; 3.8 × 10^5^/well and DU-145; 3.4 × 10^5^/well) with various concentrations (10, 20, and 40 μM) of EGCG in serum-free media was seeded on top of the polymerized matrigel for 24 h at 37 °C. For the inhibitor assay, cells were treated with 20 μM EGCG in the presence or absence of 2 μg/mL SC79 (Sigma-Aldrich) Images were captured using an inverted light microscope Ts2_PH at 40× magnification (Nikon), and the number of VM structures was quantified.

### 4.5. Western Blot Analysis

Cells (4.2 × 10^5^) were seeded in a six-well plate and treated with various concentrations (10, 20, and 40 μM) of EGCG in serum-free media for 24 h or with 20 μM EGCG for 2, 6, and 24 h. The cells were lysed in Radioimmunoprecipitation assay (RIPA) buffer containing phosphatase and protease inhibitors (Thermo Scientific, Rockford, IL). Total proteins (20 μg) from cell lysates were separated on an 8%, 10%, or 12% SDS-PAGE gel before being transferred onto a nitrocellulose transfer membrane (Pall Corporation, Port Washington, NY, USA) for 110 min at 300 mA. The membranes were soaked in 5% nonfat skim milk or 5% bovine serum albumin (BSA) for 90 min at room temperature (RT), and then probed with specific primary antibodies (Table 1) at 4 °C overnight, followed by appropriate secondary antibodies (Table 1) for 2 h at RT. Protein bands were detected using an Enhanced chemiluminescence detection kit (GE Healthcare, Chicago, IL) and quantified using an ImageJ 1.40g software (National Institute of Health, Bethesda, MD).

### 4.6. Immunofluorescence Assay

Cells (1.2 × 10^5^ cells/well) were seeded in an eight-well chamber slide and treated with 20 μM EGCG for 24 h. The cells were fixed with 3.7% formaldehyde for 10 min, permeabilized with 0.2% Triton-X 100 in phosphate-buffered saline (PBS) for 10 min, and blocked with 5% BSA in PBS for 1 h at RT. After washing three times with PBS, the cells were incubated with twist antibody (Abcam, Cambridge, UK, 1:50) overnight at 4 °C, and then goat anti-mouse IgG by Fluorescein isothiocyanate (FITC)-conjugated antibody (Millipore, Temecula, CA, 1:100) for 1 h at RT in the dark. After washing three times with PBS, the cells were counterstained with 4′,6-diamidino-2-phenylindole (DAPI, 1 μg/mL) in PBS for 5 min, mounted with 30% glycerol in PBS, and observed with a FLUOVIEW FV10i confocal microscope at 600× magnification (Olympus, Tokyo, Japan).

### 4.7. Statistical Analysis

All data were expressed as means ± SD. Results were statistically analyzed by Student’s *t*-test using Sigma plot software (Systat Software Inc., San Jose, CA, USA). A value of *p* < 0.05 was considered as statistically significant.

## Figures and Tables

**Figure 1 ijms-21-00439-f001:**
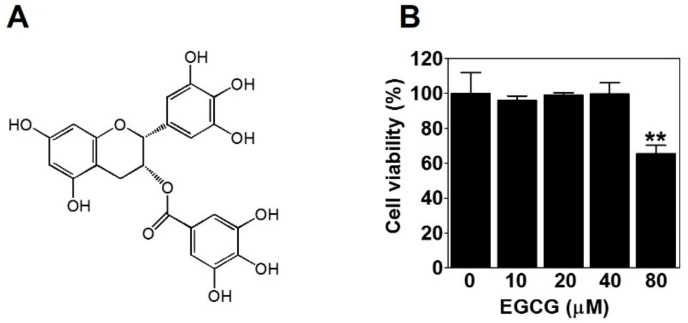
The effect of epigallocatechin-3-gallate (EGCG) on the viability of PC-3 cells. (**A**) Chemical structure of EGCG. (**B**) Cells were treated with various concentrations of EGCG (10, 20, 40, and 80 μM) for 24 h, and cell viability was determined by MTT assay. Data are expressed as means ± SD. Results were statistically calculated by Student’s *t*-test. ** *p* < 0.01 vs. untreated control.

**Figure 2 ijms-21-00439-f002:**
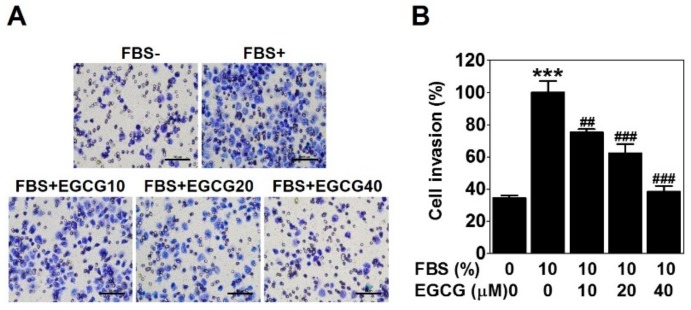
EGCG reduces the invasion of PC-3 cells. The cell invasion assay was performed using a Transwell with a matrigel-coated membrane filter for 24 h. Cells were stained, and noninvaded cells on the upper surface of the filter were removed. (**A**) Images were photographed at 200× magnification. Scale bar = 100 μm. (**B**) The number of cells invading the lower surface of the filter was quantified. Data are expressed as means ± SD. Results were statistically calculated by Student’s *t*-test. *** *p* < 0.001 vs. untreated control; ## *p* < 0.01 and ### *p* < 0.001 vs. fetal bovine serum (FBS)-treated control.

**Figure 3 ijms-21-00439-f003:**
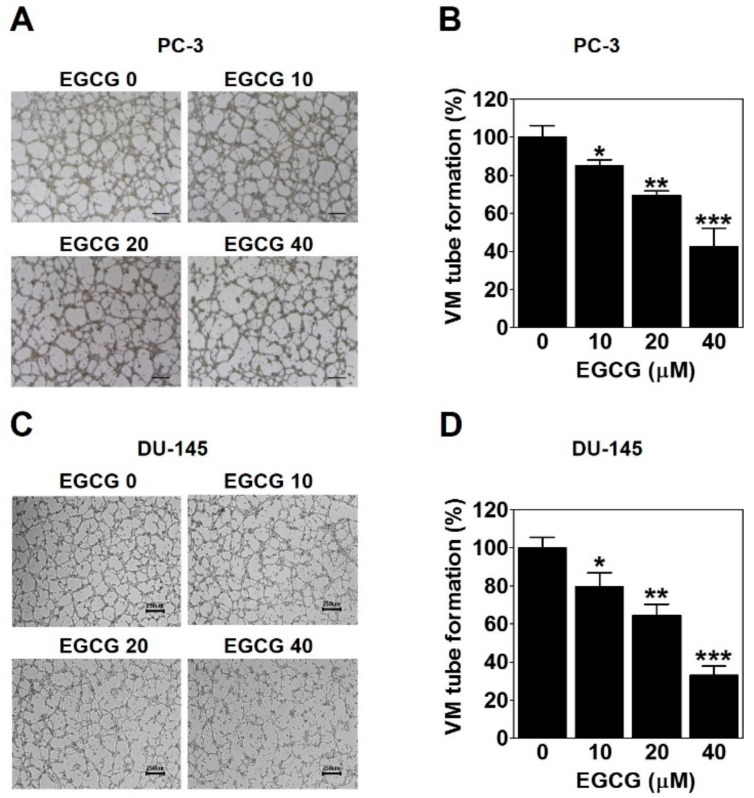
EGCG inhibits the vasculogenic mimicry (VM) of PC-3 and DU-145 cells. A cell suspension with EGCG was seeded into matrigel-coated wells and incubated for 24 h. (**A**,**C**) VM structures were photographed at 40× magnification. Scale bar = 250 μm. (**B**,**D**) The number of VM structures was quantified. Data are expressed as means ± SD. Results were statistically calculated by Student’s *t*-test. * *p* < 0.05, ** *p* < 0.01, and *** *p* < 0.001 vs. untreated control.

**Figure 4 ijms-21-00439-f004:**
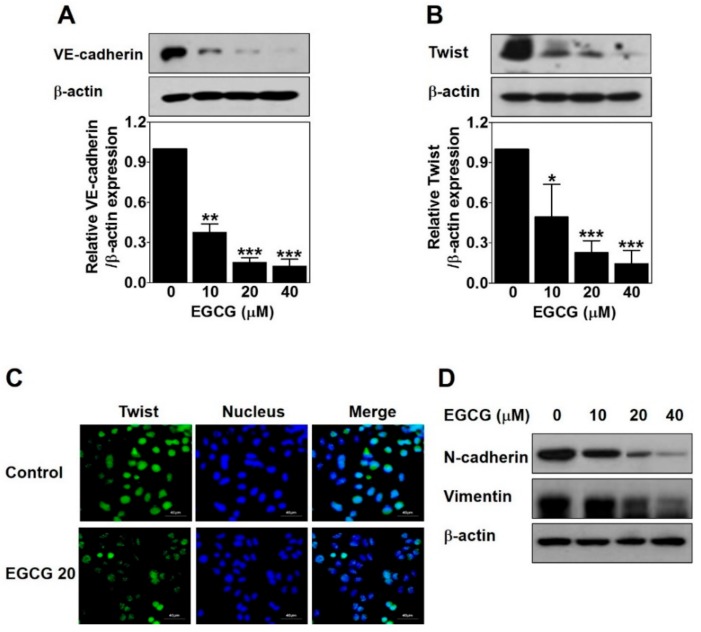
EGCG downregulates vascular endothelial cadherin (VE-cadherin) expression through inhibiting nuclear twist in PC-3 cells. Cells were treated with various concentrations of EGCG (10, 20, and 40 μM) for 24 h. (**A**,**B**) Total protein lysates (20 μg) analyzed by Western blot using specific antibodies. β-actin was used as a loading control. Data are expressed as means ± SD. Results were statistically calculated by Student’s *t*-test. * *p* < 0.05, ** *p* < 0.01, and *** *p* < 0.001 vs. untreated control. (**C**) The cells treated with or without 20 μM EGCG for 24 h were incubated with twist antibody, followed by Fluorescein isothiocyanate (FITC) conjugate secondary antibody. After counterstaining with 4′,6-diamidino-2-phenylindole (DAPI, nucleus), images were photographed at 600× magnification. Scale bar = 40 μm. (**D**) Total protein lysates (20 μg) analyzed by Western blot using specific antibodies. β-actin was used as a loading control.

**Figure 5 ijms-21-00439-f005:**
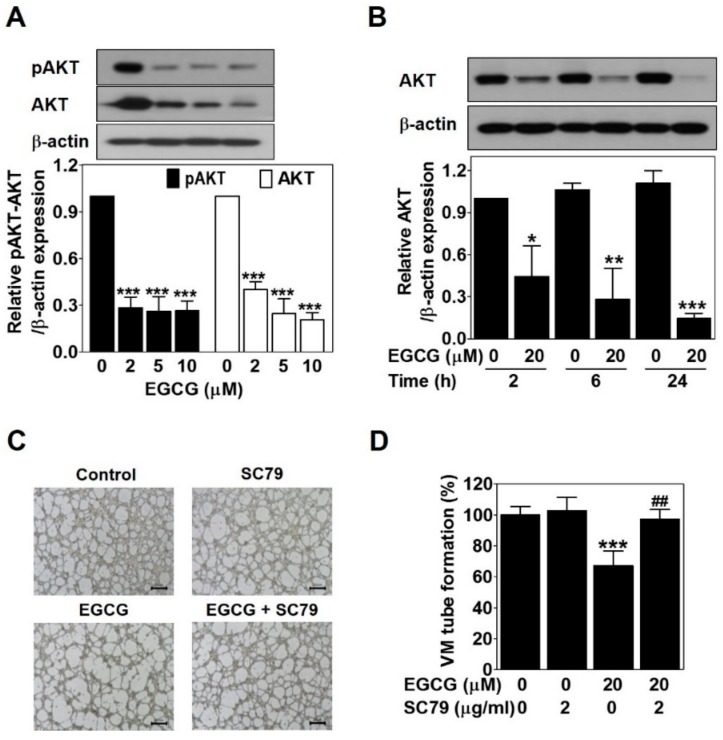
EGCG suppresses the VM of PC-3 cells through inhibiting the AKT pathway. Cells were treated with various concentrations of EGCG (10, 20, and 40 μM) for 24 h (**A**) and with 20 μM EGCG for 2, 6, and 24 h (**B**). Total protein lysates (20 μg) analyzed by Western blot using specific antibodies. β-actin was used as a loading control. Data are expressed as means ± SD. Results were statistically calculated by Student’s *t*-test. * *p* < 0.05, ** *p* < 0.01, and *** *p* < 0.001 vs. untreated control. (**C**) Cells were pretreated with 2 μg/mL SC79 for 1 h and then treated with 20 μM EGCG for 24 h. VM structures were photographed at 40× magnification. Scale bar = 250 μm. (**D**) The number of VM structures was quantified. Data are expressed as means ± SD. Results were statistically calculated by Student’s *t*-test. *** *p* < 0.001 vs. untreated control; ## *p* < 0.01 vs. only EGCG-treated control.

**Table 1 ijms-21-00439-t001:** Antibodies used in this study. VE—vascular endothelial.

Antibody	Company	Dilution	Product No.
VE-cadherin	Abgent	1:1000	AP27724a
Twist	Abcam	1:500	ab5088
β-actin	Sigma-Aldrich	1:20,000	A5316
N-cadherin	CST	1:1000	4061
Vimentin	CST	1:1000	5741
Phospho-AKT	CST	1:1000	4060
AKT	CST	1:2000	4691
goat anti-rabbit IgG-HRP	CST	1:5000	7074P2
goat anti-mouse IgG-HRP	Bio-Rad	1:5000	STAR120P

Abgent (San Diego, CA, USA); Abcam (Cambridge, UK); Sigma-Aldrich (St. Louis, MO, USA); CST, Cell Signaling Technology (Beverly, MA, USA); Bio-Rad (Langford Lane, Kidlington).

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
