# Peer review of "Epigallocatechin-3-Gallate Suppresses Vasculogenic Mimicry through Inhibiting the Twist/VE-Cadherin/AKT Pathway in Human Prostate Cancer PC-3 Cells"

_ijms, 2020, doi:10.3390/ijms21020439_

Round 1

Reviewer 1 Report

The authors present manuscript describes the efficacy of EGCG for tumor supression via VM inhibition. This study is interesting. However, their results are not conclusive for the paucity of the samples because this study uses only one cell line. The authors are requested to study using one more cell line (e.g. DU145).

Author Response

The authors present manuscript describes the efficacy of EGCG for tumor supression via VM inhibition. This study is interesting. However, their results are not conclusive for the paucity of the samples because this study uses only one cell line. The authors are requested to study using one more cell line (e.g. DU145).

(Response) Our lab have 3 types of prostate cancer cell lines, PC-3, DU-145 and LNCaP. LNCap that fails to VM. PC-3 and DU-145 are able to form VM (J Cancer. 2016 Jun 5;7(9):1114-24. doi: 10.7150/jca.14120.). Therefore, according to your comment, we have additionally performed VM formation in DU-145 cells and added these results in Fig. 3C and D in this revised MS.

Reviewer 2 Report

Steroid receptors play a pivotal role in prostate cancer (doi: 10.18632/oncotarget.6220;  doi: 10.3389/fonc.2018.00002; doi: 10.3390/cancers1110141). Epigallocatechin-3-gallate is able to modulate the expression levels of AR and ER. It could be interesting analyze the effect of Epigallocatechin-3-gallate in additional prostate cancer cell line (AR positive). PC-3 cells are AR negative but express both ERs. Did you analyzed the ERs expression level upon Epigallocatechin-3-gallate treatment?

Did the authors analysed the effect of EGCG without serum on cell invasion?

The following sentence “EGCG Downregulates VE-cadherin Expression through Inhibiting the Nuclear Twist in PC-3 Cells” is not supported by results. Authors did not analysed the nucleus- cytosol fraction by western blot analysis. Did not performed Twist silencing and etc. The effects observed in figure 4C are related to the decrease in Twist expression level.

Did the authors analysed the vimentin expression level or other markers of EMT?

The following sentence “EGCG Suppresses the VM of PC-3 Cells through Inhibiting the AKT Pathway” is not supported by results. Did the treatment affect the AKT phosphorylation? It is relevant the activity of AKT. A WB analysis of P-AKT should be included and experiments should be performed also with LY294002.

Author Response

Steroid receptors play a pivotal role in prostate cancer (doi: 10.18632/oncotarget.6220;  doi: 10.3389/fonc.2018.00002; doi: 10.3390/cancers1110141). Epigallocatechin-3-gallate is able to modulate the expression levels of AR and ER. It could be interesting analyze the effect of Epigallocatechin-3-gallate in additional prostate cancer cell line (AR positive). PC-3 cells are AR negative but express both ERs. Did you analyzed the ERs expression level upon Epigallocatechin-3-gallate treatment?

(Response) Thanks for your valuable comments. Our lab have 3 types of prostate cancer cell lines, PC-3, DU-145 and LNCaP. LNCap is AR sensitive cell line and fails to form VM. PC-3 and DU-145 are able to form VM (J Cancer. 2016 Jun 5;7(9):1114-24. doi: 10.7150/jca.14120.). Therefore, we can not compare LNCap and PC-3. Instead, we have additionally performed VM formation in DU-145 cells and added these results in Fig. 3C and D in this revised MS. As you said, PC-3 cells express ER-a and b, DU-145 and LNCaP cells expresses only ER-b. When we investigated the effect of EGCG on VM, ERs expression level was not considered. So, we did not analyze ERs expression in this study. The EGCG’s effect and its mechanism on VM needs further study. It is reported that ER b promotes VM (Oncogene volume 38, pages1225–1238(2019). Therefore, according to your comment, ER is a good target for VM-related cancer.

Did the authors analysed the effect of EGCG without serum on cell invasion?

(Response) No. EGCG is well known to inhibit invasion in several types of cancer under without chemoattractant. So, in this study, to observe more powerful effect of EGCG on invasion, we induced the invasion of PC-3 cells with serum as a chemoattractant.

The following sentence “EGCG Downregulates VE-cadherin Expression through Inhibiting the Nuclear Twist in PC-3 Cells” is not supported by results. Authors did not analysed the nucleus- cytosol fraction by western blot analysis. Did not performed Twist silencing and etc. The effects observed in figure 4C are related to the decrease in Twist expression level.

(Response) We performed WB for twist in total cell lysates. As a transcription factor for VE-cadherin, twist is a nuclear protein (BMC Cell Biol. 2009; 10: 47. doi: 10.1186/1471-2121-10-47, Cell Res. 2012 Jan;22(1):90-106. doi: 10.1038/cr.2011.144.). Therefore, twist protein expression in total cell lysates is equivalent to the amount of expression in the nucleus. Nucleus- cytosol fraction was not necessary. Also, we confirmed nuclear twist expression by IF analysis through counterstaining with DAPI that stains nuclei. According to merged figure in Fig 4C, twist is expressed in the nucleus and EGCG decreased nuclear twist expression. Taken together, we concluded that EGCG Downregulates VE-cadherin Expression through Inhibiting the Nuclear Twist.

Did the authors analysed the vimentin expression level or other markers of EMT?

(Response) According to your comment, we performed WB for vimentin and N-cadherin and added these results in Fig 4D in revised MS.

The following sentence “EGCG Suppresses the VM of PC-3 Cells through Inhibiting the AKT Pathway” is not supported by results. Did the treatment affect the AKT phosphorylation? It is relevant the activity of AKT. A WB analysis of P-AKT should be included and experiments should be performed also with LY294002.

(Response) We have performed WB analysis of pAKT and AKT simultaneously from the beginning. To observe pAKT and AKT in the same membrane, we first detected pAKT and then AKT after stripping the membrane and probing AKT antibody. Not only pAKT but also AKT were decreased by EGCG, whereas there was no effect on beta-actin expression. In the general case, there is a change in active phospho-form without affecting total form. However, our case, total AKT itself changed by EGCG. In this case, we judged that the change in pAKT was due to the change in total AKT. Therefore, we did not described pAKT results to focus on AKT change. However, according to your comment, we have added pAKT results in Fig 5A in revised MS. In our case, since pAKT and AKT were decreased, we treated with SC79, an AKT activator instead of LY294002, a PI3K inhibitor to assess the involvement of AKT and EGCG-inhibited VM formation.

Round 2

Reviewer 1 Report

The authors have satisfactorily addressed my questions.

Reviewer 2 Report

I accept the manuscript in the present form